# Metasurface-stabilized optical microcavities

Marcus Ossiander ®[1] ✉, Maryna Leonidivna Meretska ®[1], Sarah Rourke[1,2], Christina Spägele[1], Xinghui Yin[1], Ileana-Cristina Benea-Chelmus ®[1,3] & Federico Capasso ®[1] ✉

Cavities concentrate light and enhance its interaction with matter. Confining to microscopic volumes is necessary for many applications but space constraints in such cavities limit the design freedom. Here we demonstrate stable optical microcavities by counteracting the phase evolution of the cavity modes using an amorphous Silicon metasurface as cavity end mirror. Careful design allows us to limit the metasurface scattering losses at telecom wavelengths to less than 2% and using a distributed Bragg reflector as metasurface substrate ensures high reflectivity. Our demonstration experimentally achieves telecom-wavelength microcavities with quality factors of up to 4600, spectral resonance linewidths below 0.4 nm, and mode volumes below $2.7\lambda^3$. The method introduces freedom to stabilize modes with arbitrary transverse intensity profiles and to design cavity-enhanced hologram modes. Our approach introduces the nanoscopic light control capabilities of dielectric metasurfaces to cavity electrodynamics and is industrially scalable using semiconductor manufacturing processes.

Cavities can confine, shape, and enhance photons and vacuum fields and lie at the heart of achievements such as the laser[1], gravitational wave discovery[2], and quantum electrodynamics[3–5]. Many effects can be intensified with tighter confinement, thus, optical microcavities today are diversely applied in semiconductor lasers, sensing, and nonlinear optics[6]. Waveguide-based microcavities require no alignment and can often be manufactured using available on-chip photonics or fiber-optics techniques. In contrast, open microcavities offer direct access and a large tuning range for the cavity length and the resonant wavelengths.

When aiming at achieving small mode volumes in free-space optical cavities, spherical aberration-corrected end mirrors with radii of curvature on the order of a few to tens of micrometers are required. One way to manufacture such mirrors currently is to dimple optical substrates using focused-ion beam milling and then coating the dimples with distributed Bragg reflectors (DBRs)[7,8]. This process enables high-quality cavities, however, strain in the DBR coatings on curved surfaces imposes limits on the realizable phase profiles[7]. Dielectric metasurfaces can control the phase of light at the nanoscale by changing the size and shape of sub-wavelength metaatoms. They have

been previously used to stabilize microwave cavities[9], to split degenerate polarization states in cavities[10], to realize temperature sensing cavities[11], to filter color using optical cavities[12,13], to modify the output of laser cavities[14–17] and quantum cascade lasers[18], and to provide feedback for semiconductor lasers[19]. However, they have not yet been implemented to stabilize open optical microcavities. Here we demonstrate they are a mass-implementable, rapidly prototypable, and flat alternative for creating optical microcavities with large design capabilities.

## Results

### Hermite-Gaussian beams in cavities

Optical cavities have been studied extensively in literature[7,9,20,21]. When a well-defined propagation direction can be assigned to light, the paraxial wave equation applies. A set of solutions to the latter convenient for describing laser beams in free-space and cavities are Hermite-Gaussian beams. These beams' complex field evolution $u_{nm}(x,y,z)$ in cartesian coordinates (transverse directions $x, y$ and propagation direction $z$) is completely characterized by their transverse mode numbers $m, n \in \mathbb{N}^0$, their minimum beam waist $w_0$, and

[1]John A. Paulson School of Engineering and Applied Sciences, Harvard University, 29 Oxford St, Cambridge, MA 02138, USA. [2]University of Waterloo, Waterloo, ON N2L 3G1, Canada. [3]Hybrid Photonics Laboratory, École Polytechnique Fédérale de Lausanne, Lausanne CH-1015, Switzerland. ✉e-mail: mossiander@g.harvard.edu; capasso@seas.harvard.edu

their wavevector along the propagation direction $k$.

$$u_{nm}(x,y,z) = N_{nm} \frac{w_0}{w(z)} H_m\left(\sqrt{2}\frac{x}{w(z)}\right) H_n\left(\sqrt{2}\frac{y}{w(z)}\right) e^{-\frac{x^2+y^2}{w(z)^2}} e^{i\theta_{nm}(x,y,z)}$$

(1)

$$\theta_{nm}(x,y,z) = -kz - k\frac{x^2+y^2}{2R(z)} + \theta_{nm}^{Gouy}(z)$$

(2)

$$\theta_{nm}^{Gouy}(z) = (1+m+n)\tan^{-1}\left(\frac{z}{z_R}\right)$$

(3)

$$z_R = \frac{kw_0^2}{2}$$

(4)

$$w(z) = w_0\sqrt{1+\left(\frac{z}{z_R}\right)^2}$$

(5)

$$R(z) = z\left(1+\frac{z_R^2}{z^2}\right)$$

(6)

The presented form uses for brevity a mode's Rayleigh range $z_R$, beam waist $w(z)$, its radius of curvature $R(z)$, and the mode's Gouy phase $\theta_{nm}^{Gouy}$, a normalization constant $N_{nm}$, and the Hermite polynomials $H_m$ and $H_n$ of order $m$ and $n$.

Focused Hermite-Gaussian beams, when coupled to a planar cavity, suffer transverse spreading which strongly limits the achievable transmission and quality factor (Fig. 1a). Furthermore, a focused beam contains many components with non-negligible transverse wavevector. As the overall wavevector of light in vacuum is conserved, these transverse components decrease the wavevector along the propagation direction. Consequently, components with a non-negligible transverse wavevector are resonant in longer planar cavities. This causes unwanted broadening and asymmetry of the cavity resonances[22].

In the following, we will concentrate on planar-concave cavities in which the minimum waist is always located at the flat end mirror (Fig. 1b). To form a resonant mode within a cavity, the complex field must reproduce itself after one round-trip up to a real factor[23] (a complex factor indicates a stable cavity that is off resonance). This requires the round-trip phase $\phi_{RT}(x,y,L_{cav})$, which consists of the mode's propagation phase $\theta_{nm}(x,y,L_{cav})$ and the mirror reflection phases $\phi_{Mirror1/2}$, to fulfill

$$\phi_{RT}(x,y,L_{cav}) = 2\theta_{nm}(x,y,L_{cav}) + \phi_{Mirror1} + \phi_{Mirror2} = 2\pi q,$$

(7)

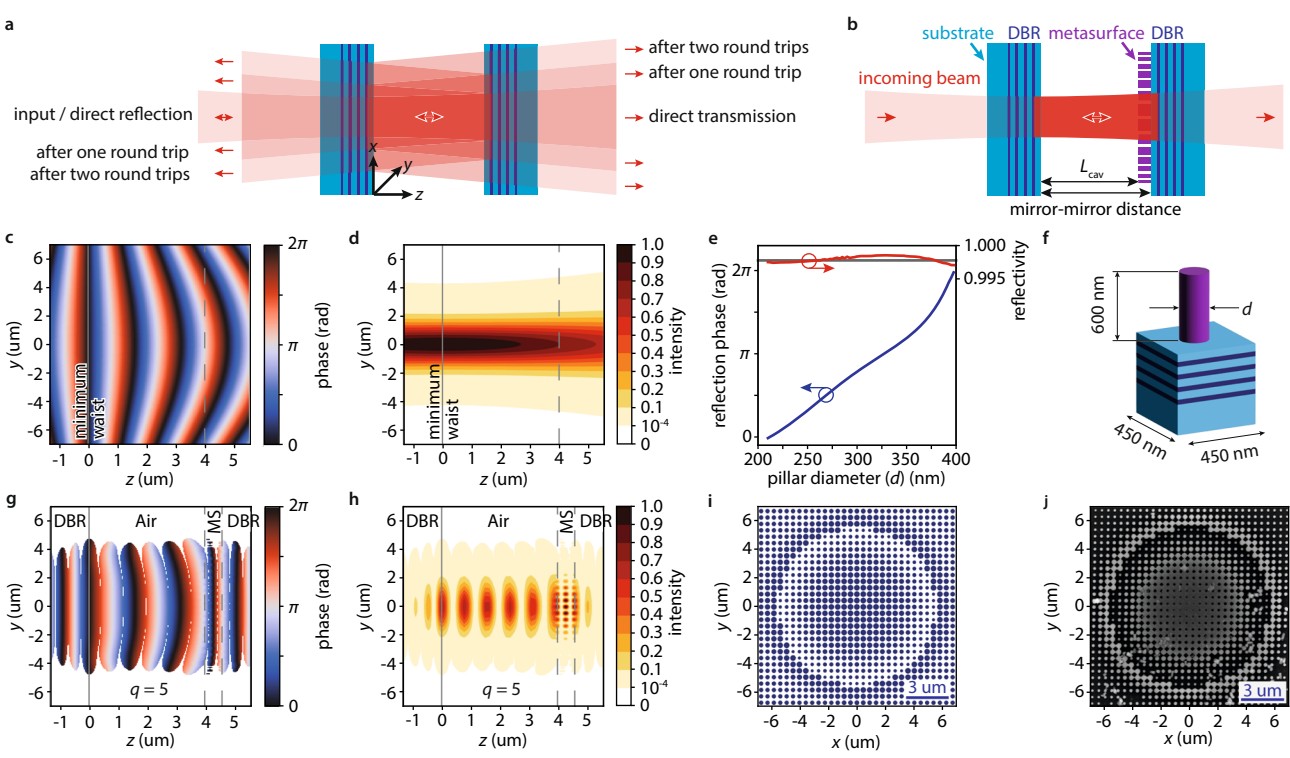

**Fig. 1 | Concept of a metasurface-stabilized optical microcavity. a** a focused Gaussian beam of light (red) coupled from the left into an optical Fabry-Perot cavity consisting of two opposing planar distributed Bragg reflectors (DBRs, light blue: Silica, dark blue: Titania) is only confined along the propagation direction (z). Transverse spreading of the beam limits field build-up in the cavity and transmission through it. **b** a metasurface (violet) placed on the surface of the second DBR matches the phase evolution of the focused Gaussian beam thus confining light also in the transverse directions (x, y). The free space cavity length $L_{cav}$ is reduced by the height of the metaatoms. **c** phase evolution (false-color plot) of a Hermite-Gaussian beam (transverse mode indices $n$, $m = 0$). Its minimum beam waist $w_0 = 1.9$ um is located at position $z = 0$ um along the propagation direction and transverse position $y = 0$ um. We design the metasurface phase to match the wavefront at $z = 4.0$ um (dashed grey line), which is equivalent to a metasurface effective radius of curvature of 20 um. **d** intensity evolution (false-color plot) of the Gaussian beam in panel (**c**). **e** metasurface nanopillar library. Simulated diameter (d) dependent reflection phase (blue line) and reflectivity (red line) of Silicon nanopillars on a Silica/Titania DBR. As a reference, the grey line shows the reflectivity of a DBR consisting of 7 Silica/Titania layer pairs without metasurface. **f** metasurface nanopillar dimensions and unit cell. Light blue: Silica, dark blue: Titania, purple: Silicon. **g** finite difference time domain modeling of the phase evolution (false color plot) of the mode with longitudinal mode index $q = 5$ in a metasurface-stabilized microcavity using the design in panel (**i**). The metasurface (MS) location is indicated by the dashed grey lines. **h** the light intensity distribution (false color plot) of the mode in panel (**g**). **i** top view of the real space metasurface design (blue: pillars). **j** scanning electron microscopy picture of the fabricated metasurface after measurement. Some metaatom pillars fell during the measurement.

where $L_{cav}$ is the cavity length and $q$ is an integer. As an example, Fig. 1c shows the phase evolution of a Hermite-Gaussian beam with $n, m = 0$; Fig. 1d shows its intensity distribution. Existing cavities[7,21] achieve transverse confinement by using a curved mirror surface that reverses the mode's propagation direction at one of the phase-isolines in Fig. 1c and thus reverses the transverse phase evolution $-k\frac{x^2+y^2}{2R(L_{cav})}$ in $\theta_{nm}(x, y, L_{cav})$. When we require that the mirror's radius of curvature matches the wavefront's radius of curvature $R_{mirror} = R(L_{cav})$ and that the Rayleigh range $z_R$ is real, we can solve for $z_R$ using Eq. (4) and find the stability criterion

$$0 \le 1 - \frac{L_{cav}}{R_{mirror}} \le 1 \qquad (8)$$

which determines for which lengths $L_{cav}$ such a cavity efficiently traps light.

## Metasurface-stabilized cavities

Metasurfaces allow to freely design an additive phase that spatially changes on the nanoscale. To realize a stable cavity, we can thus place a metasurface on the second mirror (see Fig. 1b), calculate the phase of the desired mode at its position $L_{cav}$, and design the reflection phase of the metasurface on the second DBR $\phi_{DBR+MS}(x, y)$ to reverse the wavefront evolution

$$\phi_{Mirror2}(x, y) = \phi_{DBR+MS}(x, y) = -2\theta_{nm}(x, y, L_{cav}) - \phi_{Mirror1} + 2\pi q. \quad (9)$$

This approach allows designing entirely planar stable cavities without the need for specially polished curved surfaces and can implement aspheric phase profiles without any added complexity.

To demonstrate this in practice, we choose a working wavelength $\lambda_0 = 1550$ nm due to its relevance to optical communication. We design the metasurface placed on DBRs made from alternating Silica/Titania quarter-wave layers optimized for high reflectivity (>99%) at the design wavelength. Using finite-difference-time-domain (FDTD) simulations (Lumerical Inc., FDTD), we calculate a reflection phase library for polarization-independent circular amorphous Silicon pillars. We achieve full $2\pi$ reflection phase coverage and high reflectivities for a pillar height of 600 nm, a square metaatom cell size of 450 nm, and reasonable fabrication constraints. See Fig. 1e for the pillar-diameter-dependent reflection phase and Fig. 1f for a schematic of the unit cell.

We choose an effective radius of curvature $R_{MS} = R(L_{cav}) = 20$ um for our metasurface and calculate the metasurface phase $\phi_{DBR+MS}(x, y)$ from Eq. (9). Furthermore, we fix the absolute phase offset (see below) of $\phi_{DBR+MS}(x, y)$ by choosing the cavity length $L_{cav} = 4.0$ um for the phase calculation (for Hermite-Gaussian modes, this does not mean the cavity is only resonant at this length, nor does it enhance the quality or the transmission of the longitudinal mode occurring at this length. Later in the manuscript we discuss metasurfaces realizing more complicated transverse mode profiles, for which the choice of this length is determining the length-dependent cavity transmission). This phase is designed to stabilize the fundamental transverse Hermite-Gaussian modes (transverse mode numbers $n, m = 0$) with minimum beam waists $w_0 = 1.4, 1.6, 1.8, 1.9, 2.0, \ldots$ um for the longitudinal mode numbers $q = 1, 2, 3, 4, 5, \ldots$ (see methods for longitudinal mode number counting). We then pick metaatoms from our library (Fig. 1e) to match the metasurface spatial phase. We show the final metasurface design in Fig. 1i.

The choice of pillar shape, size, and material depends on the desired function of the cavity and the intended working wavelength: here we strive for polarization independence and therefore use pillars with isotropic circular footprints. Another possible isotropic shape would be a square footprint, which behaves similarly. A polarization-dependent response, i.e., for cavity polarization filters or converters,

can be created by using anisotropic pillars with elliptical or rectangular footprint[24]. Light control in the metasurface relies on changing the reflection phase of the metasurface cavity end mirror locally, therefore we fabricate the metasurface from a material with a high refractive index and low losses at the desired wavelength. A material with these properties in the near-infrared is Silicon, which we use in this work. For applications in the visible, Titania offers low losses and a high refractive index[25], and Hafnia can be used in the ultraviolet[26]. Increasing the lateral size of the metaatoms towards the working wavelength will introduce spectral resonances. These can be used to tailor the chromatic dispersion of the metacavities[27]. Novel design techniques such as inverse design can be employed to optimize both the individual nanopillars' designs and their overall arrangement, especially if complicated functions, such as multi-wavelength behavior, are desired[28]. They can further optimize mode profiles for local field enhancement or mode volume.

## Metasurface cavity modeling

To examine our metasurface-stabilized cavity, we use this design and simulate the entire cavity using FDTD modeling (see methods). Figure 1g, h show the calculated phase evolution and intensity distribution of the $q = 5$ mode. While the wavefront curvature and transverse intensity distribution (compare with Fig. 1c, d) of the Hermite Gaussian mode is retained in the cavity, the flat metasurface planarizes the mode's wavefront within its 0.6 um pillar height by providing a phase shift $2k\frac{x^2+y^2}{2R(z)}$ which suppresses the local beam curvature. Furthermore, the intensity evolution along the propagation direction shows the expected standing wave pattern with maxima spaced by approximately $\frac{\lambda_0}{2}$. Using DBRs that are capped with the lower refractive index material (in this case Silica) locates the high-intensity anti-nodes of the cavity modes on the mirror surfaces. In a future application, this choice maximizes the interaction of the cavity mode with samples - e.g., 2d-materials or nano emitters - placed on the mirrors[29,30]. Figure 1h shows the mode's relative intensity drops below $10^{-4}$ at less than 4 um transverse distance from its center. Therefore, we expect a metasurface radius on that order sufficiently limits diffraction losses, i.e., light that is lost because it misses the metasurface due to its finite size. We then simulate the transmission of a focused beam of light through the cavity and vary the cavity length (see methods). We show results in Fig. 2a and find, as expected from Eq. (9), cavity modes spaced by approximately $\frac{\lambda_0}{2}$.

## Absolute metasurface phase effects

The absolute phase offset light experiences when passing through a metasurface is usually neglected. However, in our case, we find that it is crucial to obtaining an efficient cavity: adding an absolute offset to the metasurface phase $\phi_{DBR+MS}(x, y)$ in our cavity has two immediate consequences: the cavity lengths at which resonances appear change and the metasurface design will consist of phase-shifted metaatoms (compare the insets of Fig. 2a, b for two examples). The latter controls the zone boundaries at which the diameters of the metaatoms change abruptly due to the $2\pi$ phase jumps (see the red dashed lines in the insets in Fig. 2a, b). These abrupt changes can lead to scattering losses due to inter-element coupling (i.e., even if a small and a large adjacent pillar yield the same overall transmission phase – light coupled to those different pillars can become out of phase during its propagation along the pillars). Thus, abrupt changes should – if at all – occur in areas where the cavity mode has low intensity. The calculations in Fig. 2a, b highlight how profound the influence of these abrupt boundaries on the cavity performance can be: our simulations predict up to 50% transmission of incident light through a cavity stabilized using a metasurface with a well-chosen absolute phase. A cavity stabilized by a metasurface with the same relative phase profile but poorly chosen absolute phase achieves less than 5% transmission – a 90% efficiency loss.

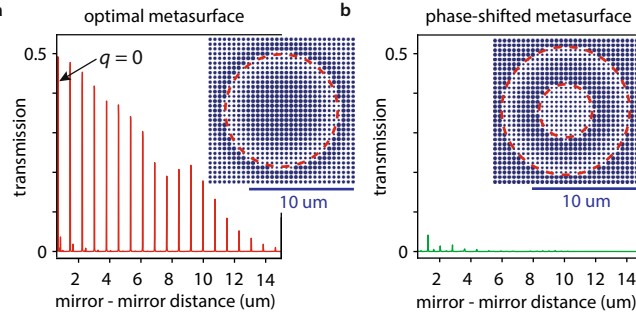

**Fig. 2 | Modeling and design of a metasurface-stabilized optical microcavity.** **a** modeled (red solid line) cavity length-dependent transmission of a metasurface-stabilized microcavity using an optimal metasurface design (see Fig. 1i). The modes with low transmission (at shorter than 5 um distance) have transverse mode numbers $n + m \geq 1$ (see also Supplementary Method 1 and Supplementary Fig. 3). The mode with the smallest longitudinal mode index $q = 0$ is marked with a black arrow. Inset: the metasurface design has one boundary at which the pillar diameter changes abruptly (red dashed line). **b** The same calculation of the cavity length-dependent transmission as in panel (**a**) for a metasurface with a poorly selected phase offset (green line). Inset: the metasurface design has two abrupt changes in the pillar diameters (red dashed lines) of which one is close to the center of the metasurface. Increased scattering at these boundaries reduces transmission through the cavity by 90%.

Due to the multiple interactions of light with the metasurface, the cavity performance depends sensitively on the metasurface's design and fabrication quality. This is further examined in Supplementary Fig. 1, which presents the effects of the fabrication accuracy and the surface roughness on the transmission properties of the cavity presented in Fig. 2a.

## Experimental results

Using top-down processing (see methods and Supplementary Fig. 2), we fabricated such metasurfaces on top of a commercially available Silica/Titania DBR terminated with a Silica layer (reflectivity >99.5%). Both, chemical and mechanical durability are experimentally verified, as in the final fabrication step (see methods), the device is immersed in bubbling Piranha solution. To increase the durability even more, i.e., for applications where the metasurface will likely touch hard objects, the Silicon pillars can be protected fully by incorporating them in a fused Silica matrix. We then placed the manufactured device opposite to a planar DBR and measured the wavelength and cavity length-dependent transmission of the resulting cavity for a focused incident light beam (numerical aperture NA ≈ 0.3, wavelength 1520 − 1580 nm, see methods for details). For the measurements, we oscillated the cavity length more than 100.000 times using a piezoelectric stage. Figure 1j shows a scanning electron microscopy picture of a final device after its measurement.

Without a metasurface (Fig. 1a), we observe broad and strongly asymmetric transmission peaks (see Fig. 3a, b, c) which is the expected behavior of a Fabry-Perot cavity. The resonance lengths, i.e., the cavity lengths at which the transmission peaks appear, shift linearly when changing the wavelength of the incident light (see Fig. 3a). For the same focusing conditions with a metasurface (Fig. 1b), we observe much narrower symmetric Lorentzian line shapes (compare Fig. 3d, e, f, with Fig. 3a, b, c), signifying a stable cavity and efficient longitudinal and transverse trapping of the incident light. A full measurement and a lineout at wavelength $\lambda = 1550$ nm are presented in Fig. 3e, f. We can see resonances with longitudinal mode numbers down to $q = 2$, indicating good parallel alignment of the two DBRs. In Fig. 4a we show the longitudinal mode index-dependent resonance transmission and length tuning bandwidth (i.e., the width of the resonance peak

indicated by the black arrows in Fig. 3f). At the working wavelength, we find finesses up to $157 \pm 7$ (see Fig. 4b), which set the upper limit for the round-trip loss to 4%, indicating less than 2% scattering losses per pass through the metasurface.

We now compare the behavior of a resonance position in the metasurface-stabilized cavity (Fig. 3d) with that of a resonance with the same longitudinal mode index $q$ in the Fabry-Perot cavity (Fig. 3a) when changing the wavelength of the incident light. The metasurface-stabilized resonance position shifts non-linearly and faster (increased slope $\frac{dL_{cav}}{d\lambda}$ in Fig. 3d). We find that the increased $\frac{dL_{cav}}{d\lambda}$ is caused by the dispersion of the metasurface, i.e., the additional group delay light experiences when it transmits through the metasurface and reflects from the metasurface cavity end mirror ($\frac{dL_{cav}}{d\lambda} = \frac{c}{\lambda}(\tau^{DBR} + \tau^{MS+DBR})$), with the speed of light $c$ and the delays $\tau^{DBR}$ and $\tau^{MS+DBR}$ caused by the penetration of light into the uncovered and metasurface-covered cavity mirrors, see methods and Refs. [12,29]). A resonance's length tuning bandwidth $\delta L_{cav}^{FWHM}$ and its spectral linewidth $\delta\lambda^{FWHM}$ are linked via $\delta L_{cav}^{FWHM} = \frac{dL_{cav}}{d\lambda}\delta\lambda^{FWHM}$. Therefore, the large slope $\frac{dL_{cav}}{d\lambda}$ we observe here decreases the spectral linewidth of the $q = 2$ longitudinal mode by more than a factor of 14 compared to a resonator without metasurface[12]. This leads to narrow spectral linewidths down to below 0.4 nm, see Fig. 4c, and highlights the application of metasurface microcavities as, e.g., narrowband spectral filters. Furthermore, this leads to large quality factors of up to $(4.6 \pm 0.4) \times 10^3$ (see Fig. 4d) at the design wavelength.

The measured cavity length-dependent resonance transmission of the Fabry-Perot cavity (Fig. 3b) shows a monotonic decrease with increasing cavity length. Conversely, the metasurface-stabilized cavity shows local dips for the longitudinal modes with the indices $q = 4$ and $q = 9, 10, 11$, see Figs. 3e, 4a. Our simulations reproduce these transmission dips for the longitudinal modes with the indices $q = 4$ and $q = 8, 9, 10$ (see Figs. 2a and 4e). We attribute the small offset of the longitudinal mode numbers to fabrication effects that cause a slightly decreased metasurface effective radius of curvature (see below). Two main factors determine the maximum transmission through the cavity for a resonant mode: the coupling of the incoming light with the cavity mode (i.e., the overlap of the incoming transverse beam profile with the cavity mode's transverse profile) and the round-trip loss of the mode itself. Whereas the first only modifies the transmission, the latter modifies the transmission and the resonance bandwidth at the same time. In our measurements, we observe the resonance linewidths increasing concurrently with the decreased transmission, see Fig. 4a, c, which identifies intra-cavity losses rooted in the finite aperture of the metasurface as the origin of the transmission dips. This is corroborated by our simulations showing that the cavity length-dependent resonance transmission behavior is independent of the incoming beam waist size and the inverse correspondence of the resonance transmission and the diffraction loss in Fig. 4e.

## Transverse confinement and modified stability criterion

We examine the transverse confinement of light in the metasurface-stabilized cavity by comparing the resonant lengths of different transverse modes (see Supplementary Method 1 and Supplementary Fig. 3). We find that the manufactured metasurface achieves a mode with the target minimum mode waist of $w_0 = (2.00 \pm 0.03)$ um close to the design cavity length $L_{cav} = 4.0$ um (mirror-mirror distance $L_{mirror-mirror} = 4.6$ um) albeit having a slightly smaller than targeted effective radius of curvature of $R_{MS} = (16.3 \pm 0.5)$ um. Furthermore, we find that the penetration of light into the planar DBR and the metasurface-covered DBR makes the cavity appear longer than $L_{cav}$ when calculating the Gouy phase and the cavity mode's radius of curvature (see Supplementary Method 1). We account for this by introducing a modal penetration depth sum[29] $L_D^{DBR} + L_D^{DBR+MS} = (2.8 \pm 0.2)$ um.

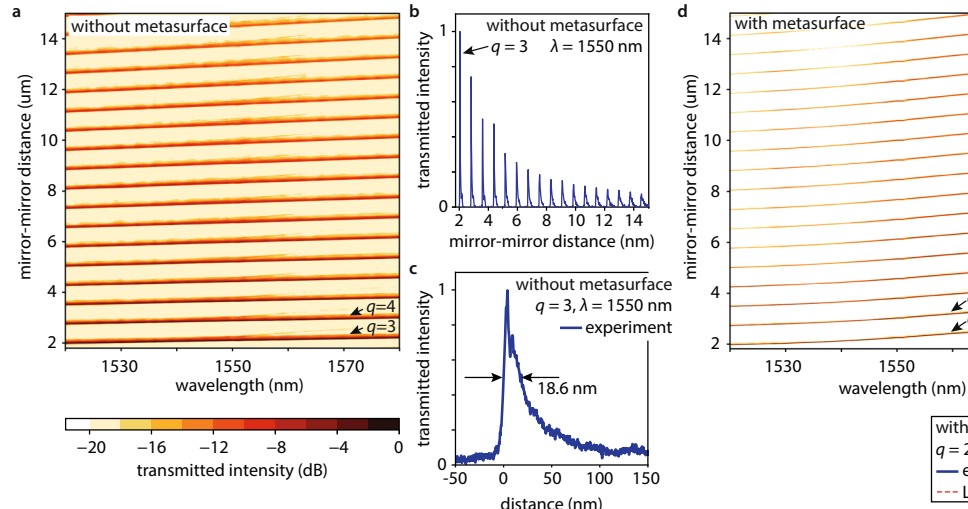

**Fig. 3 | Experimental demonstration of a metasurface-stabilized optical microcavity. a** measured wavelength- and cavity length-dependent transmission (normalized) of a laser beam focused into an unstabilized Fabry Perot cavity (false color plot). $q$ denotes the longitudinal mode index. **b** measured cavity length-dependent transmission (normalized) of a laser beam with wavelength $\lambda = 1550$ nm focused into an unstabilized Fabry Perot cavity (blue line). **c** measured cavity length-dependent transmission (normalized) of a laser beam with wavelength $\lambda = 1550$ nm focused into an unstabilized Fabry Perot cavity (blue line) in the vicinity of the resonant length of the longitudinal mode with index $q = 3$. The black arrows indicate the full width at half maximum (FWHM) of the length tuning bandwidth. **d** measured wavelength- and cavity length-dependent transmission (normalized) of

a laser beam focused into a metasurface-stabilized microcavity (false color plot). $m$ and $n$ denote the transverse mode numbers. For color bar see panel (**a**). **e** measured cavity length-dependent transmission (normalized) of a laser beam with wavelength $\lambda = 1550$ nm focused into the metasurface-stabilized microcavity (blue line). **f** measured cavity length-dependent transmission (normalized) of a laser beam with wavelength $\lambda = 1550$ nm focused into the metasurface-stabilized microcavity (blue line) in the vicinity of the resonant length of the transverse fundamental mode $(m + n = 0)$ with longitudinal mode index $q = 2$. Lorentzian fit to the data (red dashed line). The black arrows indicate the full width at half maximum (FWHM) of the length tuning bandwidth.

Because this affects the reproduction of the modes after one cavity round trip, the modal penetration depth sum modifies the cavity stability criterion. Replacing $L_{cav} \rightarrow L_{cav} + L_D^{DBR} + L_D^{DBR+MS}$ in Eq. (8) yields a modified cavity stability criterion $0 \leq 1 - \frac{L_{cav} + L_D^{DBR} + L_D^{DBR+MS}}{R_{MS}} \leq 1$. With this modified criterion, our measured effective radius of curvature and modal penetration depth sum predict cavity stability up to a longitudinal mode index $q \leq 17 \pm 1$. Indeed, we observe a steep increase of the cavity modes' bandwidths for longitudinal mode indices $q > 16$, see Fig. 4c.

**Mode volume**

As our experimental results are well reproduced by FDTD simulations, we model our cavity's mode volumes and show their evolution in Fig. 4f. For the experimental mode with longitudinal mode index $q = 2$, we find a volume of $V < 2.7\lambda^3$ ($V < 22\left(\frac{\lambda}{2}\right)^3$). This is comparable to the values reported for traditionally fabricated open access microcavities[7]. Our simulations predict that this can be reduced to $V < 1.5\lambda^3$ ($V < 12\left(\frac{\lambda}{2}\right)^3$) when decreasing the minimum cavity length (currently limited to larger than 1.5 um by technical constraints in our setup) and even further by employing high-refractive-index-terminated DBRs. The measured quality factors and calculated mode volumes suggest achieving Purcell enhancement[31,32] of 250 is already possible using this demonstrator device.

**Cavity-enhanced hologram modes**

Due to the large design freedom offered by dielectric metasurfaces, the presented concept can be adapted to create cavity modes with arbitrary transverse intensity profiles, see Fig. 5. We start with the desired mode intensity profile $I(x, y, z = 0)$ on the planar cavity mirror. We can then calculate the mode profile in the planar cavity mirror plane $u(x, y, z = 0) = \sqrt{I(x, y, z = 0)}$ and its evolution into another plane $u(x, y, z = L_{cav})$ along the propagation direction from the Rayleigh-

Sommerfeld diffraction integral[33]:

$$u(x, y, z) = -\frac{ik}{2\pi} \iint dx'dy' u(x', y', z = 0) \frac{\exp(ikr)}{r} \cos(\chi) \quad (10)$$

$$r = \sqrt{(x - x')^2 + (y - y')^2 + z^2} \quad (11)$$

$$\chi = \mathrm{atan}\left(\frac{\sqrt{(x - x')^2 + (y - y')^2}}{z}\right) \quad (12)$$

The propagation phase from the $z = 0$ to the $z = L_{cav}$ plane is given by $\arg(u(x, y, L_{cav}))$. Because a metasurface can realize arbitrary phase profiles $\phi_{DBR+MS}(x, y)$, it can readily reverse this propagation phase and stabilize such a mode in a cavity by fulfilling the modified round trip condition (compare with Eq. (9))

$$\phi_{DBR+MS}(x, y) = -2\arg(u(x, y, L_{cav})) - \phi_{Mirror1} + 2\pi q. \quad (13)$$

Even if we illuminate the resulting cavity with a beam that does not have the desired mode profile, e.g., a Gaussian beam or a plane wave, the metasurface designed using the above method will only build up the desired mode in the cavity.

To examine the viability of this approach we choose a cavity mode with an H-shaped intensity profile (see Fig. 5a) and a cavity length $L_{cav} = 10.3$ um. Using Eq. (10) and Eq. (13), we calculate $\phi_{DBR+MS}(x, y)$ and create a metasurface design by matching this phase with metaatoms. In this step, we fine-tune $L_{cav}$ so the absolute phase of the metasurface causes no abrupt pillar diameter changes in high-intensity regions of the cavity mode (see Absolute Metasurface Phase Effects).

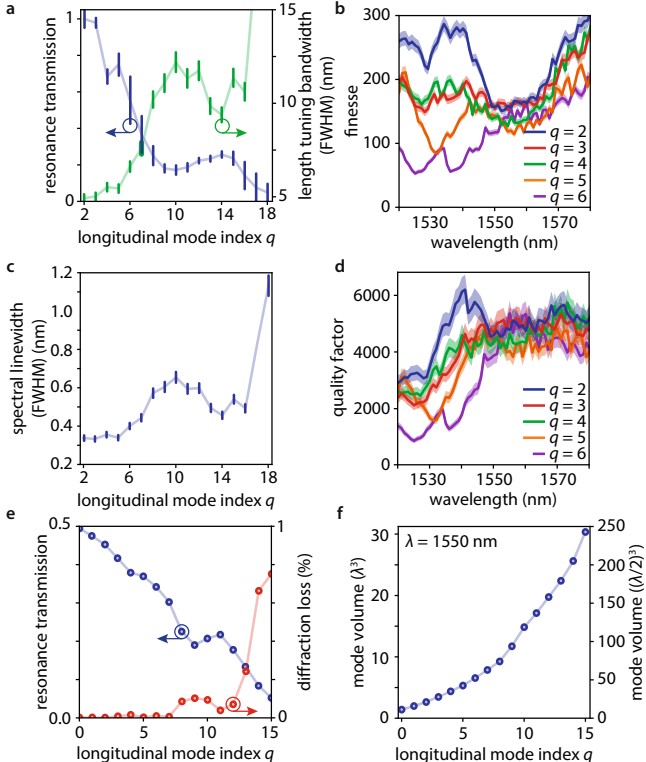

**Fig. 4 | Properties of the metasurface-stabilized optical microcavity.**
**a** measured longitudinal mode index $q$-dependent relative transmission (blue lines, i.e., the heights of the resonance peaks in Fig. 3e) and length tuning bandwidth (green lines, full width at half maximum (FWHM), i.e., the width of the resonance peak indicated by the black arrows in Fig. 3f) of the metasurface-stabilized transverse fundamental cavity modes. The error bars denote standard deviations calculated from the resonance fit and the measured piezo nonlinearity. **b** measured longitudinal mode index- and wavelength-dependent finesse of the metasurface-stabilized transverse fundamental cavity modes (colored lines). Shaded areas denote standard deviations calculated from the resonance fit and the measured piezo nonlinearity. **c** measured longitudinal mode index-dependent spectral linewidth (blue lines), of the metasurface-stabilized transverse fundamental mode cavity modes. The error bars denote standard deviations calculated from the resonance fit and the measured nonlinearity of the piezoelectric stage. **d** measured longitudinal mode index- and wavelength-dependent quality factor of the metasurface-stabilized cavity modes with $n + m = 0$ (colored lines). Shaded areas denote standard deviations calculated from the resonance fit and the measured nonlinearity of the piezoelectric stage. **e** calculated longitudinal mode index-dependent resonance transmission (blue dots and line, i.e., the evolution of the resonance peak heights in Fig. 2a) and the round-trip diffraction losses (red dots and line) for the optimal design in Fig. 1i. **f** finite difference time domain modeling of the longitudinal mode index-dependent mode volumes (blue dots and line) for the metasurface-stabilized microcavity using the optimal design in Fig. 1i. The mode volume is measured relative to the wavelength $\lambda$.

We expect scattering losses due to sharp features in our desired mode profile (Fig. 5a), therefore we reduce the reflectivity of our DBRs and metasurface to 95% by decreasing the number of Silica/Titania layers. Figure 5c displays the final metasurface design.

We then perform FDTD simulations of an entire cavity consisting of a planar DBR on one side and an opposing DBR with the metasurface stabilizer placed on it (see Fig. 5b). Figure 5d shows the cavity mode intensity profile at the mirror without metasurface. Even though we illuminate with a Gaussian beam (beam waist radius: 7 um), the metasurface stabilizes and enhances only the desired mode profile, therefore the intra-cavity intensity has the desired H-shape with sharp edges. We find an average electric field enhancement of 6 in the desired mode compared to an incoming plane wave.

To highlight that this approach is not limited to binary or symmetric mode profiles, we choose an asymmetric and greyscale dot-pattern (see Fig. 5i). Our simulations show that a metasurface cavity designed following the previously detailed method (see Fig. 5j for the final design), illuminated with an incoming plane wave, also enhances the desired asymmetric dot pattern (see Fig. 5k).

For Hermite-Gaussian modes, a single metasurface design with a fixed radius of curvature can stabilize many longitudinal modes (with the cavity length limited by the stability criterion). This works because a mode's propagation length-dependent radius of curvature can be counteracted by a change in its minimum beam waist, such that its wavefront fits the metasurface's effective radius of curvature.

This does not apply when the desired mode is a hologram. In this situation, transmission peaks for cavity lengths that are much shorter or much longer than the design value show asymmetric mode profiles, as is characteristic of unstable modes. The transmission peak close to the design cavity length shows a Lorentzian profile (see Fig. 5h). This is further illustrated by the transverse mode profiles at resonance: close to the design cavity length (see Fig. 5d), the desired H-shape is produced. Off of the design cavity length (see inset in Fig. 5e), the mode profile does not resemble the desired H-shape. Another indication of the length-dependent stability is that the cavity length-dependent resonance transmission (see Fig. 5e, f) shows a local enhancement close to the design cavity length. An empty Fabry-Perot cavity, even if excited with an H-shaped mode, does not show a local enhancement at the design cavity length (see Fig. 5g), nor does the corresponding enhanced field have an H-shaped profile (see inset in Fig. 5g). This reiterates the necessity of metasurfaces to enhance modes with complicated transverse profiles. Whereas the metasurface guarantees enhancement of the longitudinal mode at the design cavity length, it is not designed to suppress other longitudinal modes. Therefore, other local maxima, especially for short cavity lengths that limit the transverse mode spreading per cavity round trip, can occur (see, e.g., Fig. 5e black dashed arrow), albeit with uncontrolled transverse mode profiles.

In summary, we combined commercially available DBRs with metasurfaces to realize stable microcavities with classical and designed mode profiles. The approach offers design freedom, is rapidly prototypable, and at the same time directly manufacturable on the industrial scale as it is fully compatible with widely available semiconductor fabrication techniques. Especially the ability to implement complicated phase profiles, design chromaticity and achromaticity[27,34], and control the polarization state[24] of light down to the ultraviolet spectral region[35] will offer better control of light in microcavities.

## Methods
### Longitudinal mode counting
We use the following convention when assigning the longitudinal mode index $q$: our DBRs are terminated by a low-refractive-index material, therefore, the cavity mode has intensity maxima at the mirror facets. Therefore, the Fabry-Perot cavity without metasurface supports a cavity mode at $L_{cav} \approx 0$ um (because of the light penetration into the DBRs), to which we assign the mode number $q = 0$. Accordingly, the longitudinal mode with index $q = 1$ occurs at a free space cavity length $L_{cav} \approx \frac{\lambda}{2}$ and so on. When comparing the Fabry-Perot cavity with the metasurface-stabilized cavity, at the same distance between the DBRs $L_{mirror-mirror}$, the metasurface-stabilized cavity length is 0.6 um smaller because the metasurface is 0.6 um high (compare Fig. 1b and Supplementary Fig. 4). Therefore, at the same $L_{mirror-mirror}$, the longitudinal mode numbers assigned to modes in the metasurface-stabilized cavity are roughly one smaller than modes that occur in the Fabry-Perot cavity.

### Fabrication
The fabrication process is summarized in Supplementary Fig. 2. Using plasma-enhanced chemical vapor deposition, we deposit a 600 nm-

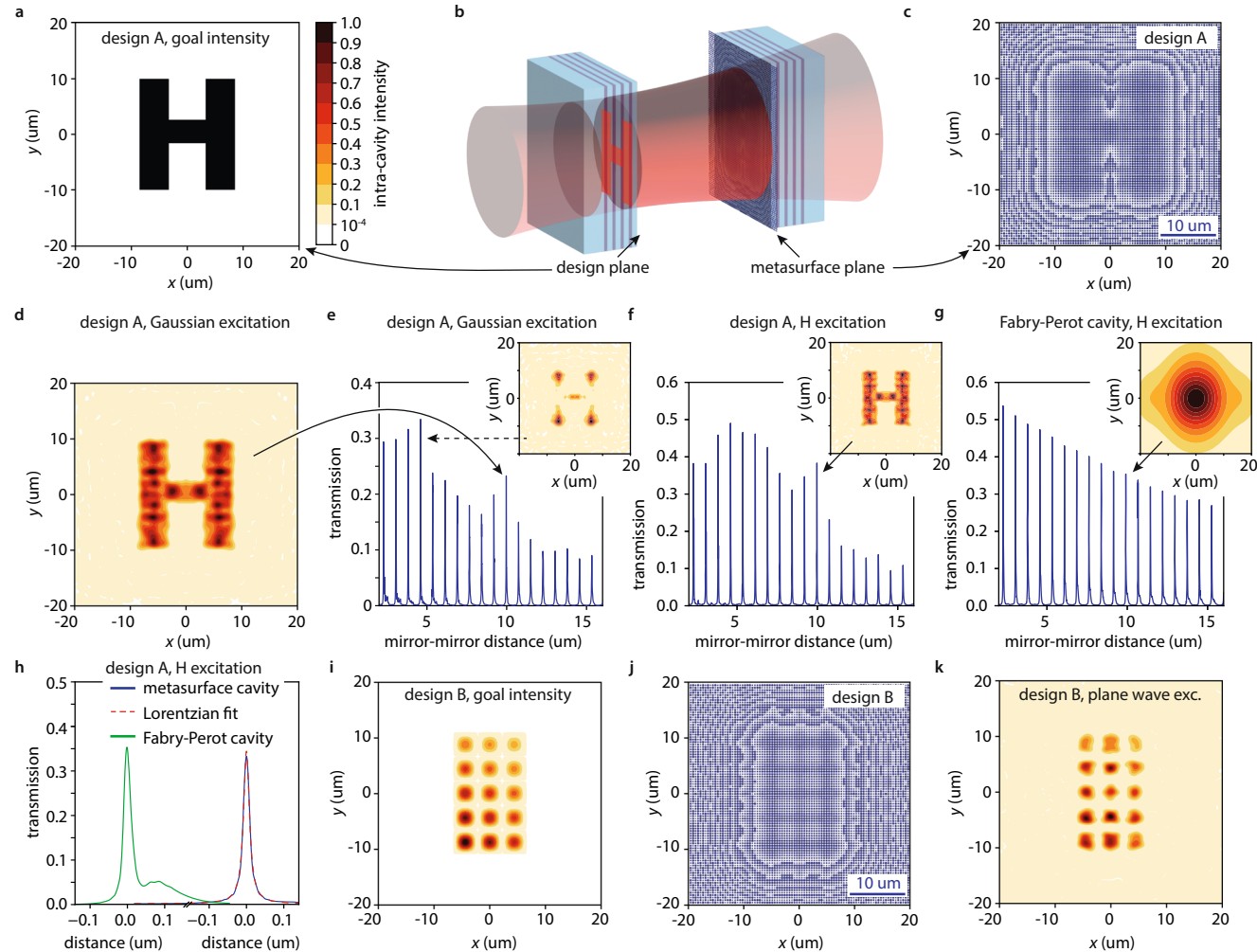

**Fig. 5 | Metasurface microcavities with designer mode profiles. a** desired transverse cavity mode intensity profile in the design plane (design A, the color bar is valid for all panels). $x, y$ are the transverse spatial coordinates. **b** cavity setup: a light beam (red) is coupled to a cavity through a planar distributed Bragg reflector (DBR). A metasurface on the opposing DBR (labeled metasurface plane) is designed to satisfy the cavity round trip condition for a chosen intensity profile (red H-shape) in the design plane. **c** top view of the calculated metasurface design (design A, blue: pillars). **d** modeled mode profile in the design plane of a metasurface microcavity using design A illuminated with a Gaussian beam. Only the designed H-profile is enhanced. **e** modeled (blue line) cavity length-dependent transmission of a metasurface microcavity using a design A metasurface excited by a Gaussian beam. We observe a local maximum of the transmission peak (solid black arrow) at the design length (mode profile in panel (**d**)). Inset: the mode profile far from the design length

(black dashed arrow) does not reproduce the H-shape. **f** same as panel (**e**) but excited by an H-shaped beam (transverse profile in panel (**a**)). We observe a local maximum of the transmission peak (solid black arrow) at the design length. Inset: the H-shaped cavity mode profile at the design length. **g** modeled (blue line) transmission of an unstable Fabry-Perot cavity excited by the same H-shaped beam. Inset: the mode profile in the Fabry-Perot cavity at the design length of the H-cavity (solid black arrow) does not resemble the H-shape. **h** magnified transmission of the resonances marked with the arrows in panels (**f**) and (**g**) (solid blue and green lines) and Lorentzian fit (red dashed line). The Fabry-Perot cavity is unstable (asymmetric resonance shape) whereas the metasurface cavity is stable (Lorentzian resonance shape). **i** a desired asymmetric dot pattern intensity distribution (design B). **j** top view of the metasurface design realizing design B (blue: pillars). **k** modeled mode profile realized with a design B metasurface cavity illuminated by a plane wave.

thick amorphous Silicon layer on commercially available DBRs (Eksma 031-1550-i0). On top, we spin-coat a layer of negative electron beam resist (Micro Resist Technology ma-N 2403) and subsequently a conductive polymer (Showa Denko ESPACER 300) to avoid charging effects. We then write the metasurface mask patterns using electron beam lithography (Elionix HS-50). After developing, (MicroChemicals AZ726 MIF) we remove amorphous Silicon in non-exposed areas using inductively coupled plasma-reactive ion etching (ICP-RIE using $SF_6$ and $C_4F_8$). Finally, we remove the remaining electron beam resist using Piranha solution.

### Experimental setup

To characterize our metasurface-stabilized cavities, we use coherent light from a tunable semiconductor laser (Santec TSL-550), which we focus using an aspheric lens (NA ≈ 0.3) through the 3 mm thick

substrate of a planar DBR mirror (Eksma 031-1550-i0, reflectivity >99.5% for wavelengths between 1520 nm and 1570 nm). Excluding a 2 × 2 mm area in its center, we grind down the front facet of this DBR to allow small cavity lengths even for imperfect angle alignment regarding a second, opposing, planar DBR. This second DBR has the metasurface on its surface and is mounted on a three-axis stage (Thorlabs NanoMax). This mirror order avoids changing the position of the incoming beam waist with respect to the beam waist of the cavity modes when varying the cavity length with the stage's piezo actuators. Transmitted light is collected using a lens (NA ≈ 0.3) and detected using an amplified InGaAs detector (Thorlabs PDA10CS2).

### Length calibration

To calibrate the length of our microcavity, we move the minimum beam waist of the incoming beam only along the propagation

direction. The resulting increase of the beam waist on the planar cavity end mirror leads to strong excitation of the planar-planar Fabry-Perot cavity modes around the metasurface, shown by their asymmetric resonance profile, see Fig. 3a, b. The slope $\frac{dL_{mirror-mirror}}{d\lambda}$ then reveals the sum of the current cavity length $L_{mirror-mirror}$ and the frequency penetration depth into the two planar DBRs $L_\tau^{DBR}$ ($\frac{L_\tau^{DBR}}{c} = \tau^{DBR}$)

$$\lambda \frac{dL_{mirror-mirror}}{d\lambda} = L_{mirror-mirror} + L_\tau^{DBR} + L_\tau^{DBR}. \tag{14}$$

To correct for the DBR penetration, we analytically calculate and simulate (see below) $L_\tau^{DBR} = 1.5$ um. Results from both methods coincide. We then subtract it from $\lambda \frac{dL_{mirror-mirror}}{d\lambda}$ to obtain $L_{mirror-mirror}$. The metasurface height is 0.6 um, thus the length of the stabilized cavity is $L_{cav} = L_{mirror-mirror} - 0.6$ um. Furthermore, as the metasurface height is constant, $\frac{dL_{mirror-mirror}}{d\lambda} = \frac{dL_{cav}}{d\lambda}$.

### Calculation of the finesse and the quality factor

We determine the finesse $F \approx \frac{2\pi}{\delta\theta^{FWHM}} = \frac{2\pi}{2k\delta L_{cav}^{FWHM}} = \frac{\lambda}{2\delta L_{cav}^{FWHM}}$ from the full-width-at-half-maximum length tuning bandwidth $\delta L_{cav}^{FWHM}$, an expression which we derived from the full-width-at-half-maximum phase linewidth $\delta\theta^{FWHM} = 2k\delta L_{cav}^{FWHM}$, see Refs. [7,21]. Subsequently, we calculate the quality factor using $Q = \frac{\omega}{\delta\omega^{FWHM}} \approx \frac{\lambda}{\delta\lambda^{FWHM}} = \frac{L_{cav} + L_\tau^{DBR} + L_\tau^{DBR+MS}}{\delta L_{cav}^{FWHM}}$ (using the angular frequency $\omega$, the angular frequency spectral linewidth $\delta\omega^{FWHM}$, and the wavelength spectral linewidth $\delta\lambda^{FWHM}$). As the quality factor measures the dissipated energy per oscillation period, we use the sum of the cavity length $L_{cav}$ and the frequency-penetration-depths $L_\tau^{DBR} + L_\tau^{DBR+MS}$ into the cavity mirrors[7,21,29] ($\frac{L_\tau^{DBR} + L_\tau^{DBR+MS}}{c} = \tau^{DBR} + \tau^{DBR+MS}$). To determine $L_{cav} + L_\tau^{DBR} + L_\tau^{DBR+MS}$, we measure the slope of the resonance condition $\frac{dL_{mirror-mirror}}{d\lambda}$ and use the modified Eq. (14):

$$\lambda \frac{dL_{mirror-mirror}}{d\lambda} = L_{cav} + L_\tau^{DBR} + L_\tau^{DBR+MS}. \tag{15}$$

### Finite difference time domain simulations

A full length ($L_{cav} = 0 - 15$ um) and nanometer-resolution sweep would require excessive computational resources. To remedy that, we again use Eq. (15) to project the wavelength-dependent results of our simulations in a 10 nm bandwidth around the working wavelength on the distance axis. We determine the frequency-penetration-depths of our DBR $L_\tau^{DBR} = 1.5$ um and the metasurface-covered DBR $L_\tau^{DBR+MS} = 7.0$ um by comparing their simulated reflection phases to that of a perfect electrical conductor placed at the front facet of the DBR/metasurface. The mapping allows us to cover the entire distance sweep in 250 simulations.

### Reporting summary

Further information on research design is available in the Nature Portfolio Reporting Summary linked to this article.

## Data availability

The data supporting the findings of this study are available in figshare with the identifier https://doi.org/10.6084/m9.figshare.21923760 and from the corresponding author upon request.

## Code availability

The findings of this paper do not rely on unpublished algorithms. Design and data analysis scripts are available from the corresponding author upon request.

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

## Acknowledgements

This work was performed, in part, at the Center for Nanoscale Systems (CNS), a member of the National Nanotechnology Coordinated Infrastructure (NNCI), which is supported by the NSF under award no. ECCS-2025158. CNS is a part of Harvard University. The computations in this paper were run on the FASRC Cannon cluster supported by the FAS Division of Science Research Computing Group at Harvard University. M.O. acknowledges a Feodor-Lynen Fellowship from the Alexander von Humboldt Foundation. I.C.B.C. acknowledges support from the Swiss national science foundation through grant 181935 and from the Hans-Eggenberger foundation. F.C. acknowledges that this material is based upon research supported by the Office of Naval Research under Award Number N00014-20-1-2450 and by the Air Force Office of Scientific Research under Award Number FA9550-21-1-0312.

## Author contributions

M.O. developed the project. M.L.M. fabricated the samples. M.O., S.R., C.S., X.Y., I.C. B.C. experimented and analyzed experimental data. M.O. and F.C. wrote the manuscript. All authors discussed the final version of the manuscript.

## Competing interests

The authors declare no competing interests.
