## [Peer Review File · Nature Communications]

Metasurface-Stabilized Optical MicrocavitiesReviewer #1 (Remarks to the Author):

This manuscript intends to report stable microcavities with classical and designed profiles by combining commercially available DBRs with metasurface. It is noted that this manuscript is focused on design but lacks fundamental studies and insights. For example, the manuscript did not discuss the underlying parameters that can influence the design and performance, and just presented some analyses. So, it is difficult to justify any possibilities of publishing at research-oriented journals such as Nature Communications.

Reviewer #2 (Remarks to the Author):

The manuscript with ID NCOMMS-22-32850 by Ossiander and colleagues reports an interesting and beautifully executed study demonstrating how phase evolution of cavity modes within an open Fabry–Perot microcavity can be stabilised by generating a metasurface–featuring amorphous silicon nanopillars–on one of the highly reflecting DBR mirrors of the cavity. The study is comprehensive with robust theoretical and experimental frameworks supporting the conclusions drawn from their observations. By rationally designing the features of the metasurface, the Authors demonstrate quality enhancements in light confinement within the “conventional” cavity structure. The reported figures of merit (e.g., quality, finesse, and linewidth) are outstanding when these are compared to the same cavity structure without metasurface. The Authors also demonstrate proof-of-concept applicability of the reported structures, making the study complete. Overall, it is my opinion that the study is of the highest quality standards in the field and might also be of interests for a broad audience working in other related fields. Therefore, I strongly believe that the manuscript should be considered for publication in Nature Communications. I just have a minor suggestion for the Authors consideration: I wonder how important the features of the metasurface in terms of enhancements are. The Authors do demonstrate this using silicon pillars but have they considered other feature designs and how these could be harnessed to further maximise the performance of the system/architecture?

Reviewer #3 (Remarks to the Author):

In this paper, the authors theoretically proposed and experimentally demonstrated stable microcavities that comprise commercially available DBRs and metasurfaces. Owing to the distributed Bragg reflector and careful design, the proposed metasurface has low scattering losses and high reflectivity at telecom wavelengths. The manuscript is written well and organized nicely. Given these important factors, I think the manuscript can be published in Nature Communications after addressing the following minor comments:

- 1. In Fig. 2a, why there exists several modes with low transmission when the mirror – mirror distance ranges from 0 to 4 μm ?**
- 2. After introducing metasurface, the local enhancement of the longitudinal mode occurs at several points not only close to the design cavity length, as shown in Fig. 3f. This phenomenon is different from the statement in the manuscript. Could the authors clarify on this a bit more?**
- 3. In Fig. 4e, the first five modes cannot exhibit symmetric Lorentzian line shapes, which is similar with that without metasurface (Fig. 3c). It seems that the introduced metasurface does not work. Can the author explain it in detail and give the cavity length-dependent transmission of a microcavity without metasurface?**
- 4. It would be great if the metasurface design of the asymmetric dot-pattern is given.**

Reviewer #4 (Remarks to the Author):

Reviewer Report on Manuscript entitled "Metasurface-Stabilized Optical Microcavities" by Dr Ossiander and colleagues (Manuscript Number: NCOMMS-22-32850), to be published in Nature Communications

The manuscript is dealing with the experimental results of the stabilized optical microcavities whose one commercially available DBR mirror's surface is modified with metasurface of nano pillars. The idea and results are interesting and have significant potentials for real usage. The durability for real usage is also interesting. If authors have any knowledge for this question, please give us authors' comments. The fabrication method for metasurface is presented in Methods part, but no illustration. Some illustration for fabrication of the metasurface on DBR mirror is very useful for readers to understand how we fabricate the metasurface on DBR mirrors. The content is worth publishing in Nature Communications.

Specific comments and answers to the reviewers

In the following, we give a one-to-one response (printed in blue) to the individual reviewer's comments which are printed verbatim and in their entirety in green.

Reviewer #1 (Remarks to the Author):

This manuscript intends to report stable microcavities with classical and designed profiles by combining commercially available DBRs with metasurface. It is noted that this manuscript is focused on design but lacks fundamental studies and insights. For example, the manuscript did not discuss the underlying parameters that can influence the design and performance, and just presented some analyses. So, it is difficult to justify any possibilities of publishing at research-oriented journals such as Nature Communications.

The manuscript introduces the theory and design principles for metasurface stabilized microcavities at optical frequencies. We then present the metasurface design process, fabrication details, and experimental measurements of a real device. The theory is then refined using the results of the measurements. Finally, holographic microcavities, impossible to realize with classic optical means, are presented.

The manuscript contains extensive discussions of parameters such as the metasurface's effective radius of curvature, absolute phase, the frequency and modal penetration depth, and the cavity length. We have, furthermore, as per request of reviewer #1 and #2, added an investigation into the effect of metasurface fabrication parameters such as fabrication accuracy and surface roughness as Supplementary Fig. 4, together with a remark in the main text:

Due to the multiple interactions of light with the metasurface, the cavity performance depends sensitively on the metasurface's design and fabrication quality. This is further examined in Supplementary Fig. 4, which presents the effects of the fabrication accuracy and the surface roughness on the transmission properties of the cavity presented in Fig. 2a.

Moreover, we have added a discussion of how other design parameters can be adapted to realize microcavities with different functions or at different wavelengths:

The choice of pillar shape, size, and material depends on the desired function of the cavity and the intended working wavelength: here we strive for polarization independence and therefore use pillars with isotropic circular footprints. Another possible isotropic shape would be a square footprint, which behaves similarly. A polarization-dependent response, i.e., for cavity polarization filters or converters, can be created by using anisotropic pillars with elliptical or rectangular footprint²⁴. Light control in the metasurface relies on changing the reflection phase of the metasurface cavity end mirror locally, therefore we fabricate the metasurface from a material with a high refractive index and low losses at the desired wavelength. A material with these properties in the near-infrared is Silicon, which we use in this work. For applications in the visible, Titania offers low losses and a high refractive index²⁵, and Hafnia can be used in the ultraviolet²⁶. Increasing the lateral size of the metaatoms towards the working wavelength will introduce spectral resonances. These can be used to tailor the chromatic dispersion of the metacavities²⁷. Novel design techniques such as inverse design can be employed to optimize both the individual nanopillars' designs and their overall arrangement, especially if complicated functions, such as multi-wavelength behavior, are desired²⁸. They can further optimize mode profiles for local field enhancement or mode volume.

On these grounds and backed by the very positive reviews of reviewers #2, #3, and #4, all of which suggest publication in Nature Communications, we strongly disagree with the reviewer's assessment.

Reviewer #2 (Remarks to the Author):

The manuscript with ID NCOMMS-22-32850 by Ossiander and colleagues reports an interesting and beautifully executed study demonstrating how phase evolution of cavity modes within an open Fabry–Perot microcavity can be stabilised by generating a metasurface—featuring amorphous silicon nanopillars—on one of the highly reflecting DBR mirrors of the cavity. The study is comprehensive with robust theoretical and experimental frameworks supporting the conclusions drawn from their observations. By rationally designing the features of the metasurface, the Authors demonstrate quality enhancements in light confinement within the “conventional” cavity structure. The reported figures of merit (e.g., quality, finesse, and linewidth) are outstanding when these are compared to the same cavity structure without metasurface. The Authors also demonstrate proof-of-concept applicability of the reported structures, making the study complete. Overall, it is my opinion that the study is of the highest quality standards in the field and might also be of interests for a broad audience working in other related fields. Therefore, I strongly believe that the manuscript should be considered for publication in Nature Communications.

We thank the reviewer for the very positive review and the suggestion for publication in Nature Communications.

I just have a minor suggestion for the Authors consideration: I wonder how important the features of the metasurface in terms of enhancements are.

We have included a new supplementary figure 4 to discuss the influence of the trueness of the metasurface’s features, i.e., fabrication accuracy and surface roughness, on the cavity performance. Furthermore, we have added a reference to the main text:

Due to the multiple interactions of light with the metasurface, the cavity performance depends sensitively on the metasurface’s design and fabrication quality. This is further examined in Supplementary Fig. 4, which presents the effects of the fabrication accuracy and the surface roughness on the transmission properties of the cavity presented in Fig. 2a.

The Authors do demonstrate this using silicon pillars but have they considered other feature designs and how these could be harnessed to further maximise the performance of the system/architecture?

We have included a new paragraph introducing possible design choices for advanced applications of metasurface cavities outside the scope of the current manuscript:

The choice of pillar shape, size, and material depends on the desired function of the cavity and the intended working wavelength: here we strive for polarization independence and therefore use pillars with isotropic circular footprints. Another possible isotropic shape would be a square footprint, which behaves similarly. A polarization-dependent response, i.e., for cavity polarization filters or converters, can be created by using anisotropic pillars with elliptical or rectangular footprint²⁴. Light control in the metasurface relies on changing the reflection phase of the metasurface cavity end mirror locally, therefore we fabricate the metasurface from a material with a high refractive index and low losses at the desired wavelength. A material with these properties in the near-infrared is Silicon, which we use in this work. For applications in the visible, Titania offers low losses and a high refractive index²⁵, and Hafnia can be used in the ultraviolet²⁶. Increasing the lateral size of the metaatoms towards the working wavelength will introduce spectral resonances. These can be used to tailor the chromatic dispersion of the metacavities²⁷. Novel design techniques such as inverse design can be employed to optimize both the individual nanopillars’ designs and their overall arrangement, especially if complicated functions, such as multi-wavelength behavior, are desired²⁸. They can further optimize mode profiles for local field enhancement or mode volume.

Reviewer #3 (Remarks to the Author):

In this paper, the authors theoretically proposed and experimentally demonstrated stable microcavities that comprise commercially available DBRs and metasurfaces. Owing to the distributed Bragg reflector and careful design, the proposed metasurface has low scattering losses and high reflectivity at telecom wavelengths. The manuscript is written well and organized nicely. Given these important factors, I think the manuscript can be published in Nature Communications after addressing the following minor comments:

We thank the reviewer for the very positive review and the suggestion for publication in Nature Communications.

(1) In Fig. 2a, why there exists several modes with low transmission when the mirror – mirror distance ranges from 0 to 4 μm ?

The modes with the low transmission in Figure 2a are higher-order transverse modes, i.e., modes with $n + m \geq 1$. We can also observe these modes in the experiment, see Supplementary Fig. 2, and use them to explore the spatial confinement in our cavities. More details are given in Supplementary Method 1. To clarify and highlight this, we added a sentence to the caption of Figure 2:

The modes with low transmission (at shorter than 5 μm distance) have transverse mode numbers $n + m \geq 1$ (see also Supplementary Method 1 and Supplementary Fig. 2).

(2) After introducing metasurface, the local enhancement of the longitudinal mode occurs at several points not only close to the design cavity length, as shown in Fig. 3f. This phenomenon is different from the statement in the manuscript. Could the authors clarify on this a bit more?

We believe the reviewer applied the discussion of the metasurfaces realizing complicated mode profiles to the section describing metasurfaces that realize Hermite-Gaussian modes. We have added an extensive discussion of the local enhancements of the longitudinal modes for complicated mode profiles, see the next question below for a detailed description of the changes. To better distinguish the two cases for the readers, we have added a remark to the discussion of the metasurface realizing Hermite-Gaussian beams:

Furthermore, we fix the absolute phase offset (see below) of $\phi_{\text{DBR+MS}}(x, y)$ by choosing the cavity length $L_{\text{cav}} = 4.0 \mu\text{m}$ for the phase calculation (for Hermite-Gaussian modes, this does not mean the cavity is only resonant at this length, nor does it enhance the quality or the transmission of the longitudinal mode occurring at this length. Later in the manuscript we discuss metasurfaces realizing more complicated transverse mode profiles, for which the choice of this length is determining the length-dependent cavity transmission.). This phase is designed to stabilize the fundamental transverse Hermite-Gaussian modes (transverse mode numbers $n, m = 0$) with minimum beam waists $w_0 = 1.4, 1.6, 1.8, 1.9, 2.0, \dots \mu\text{m}$ for the longitudinal mode numbers $q = 1, 2, 3, 4, 5, \dots$ (see methods for longitudinal mode number counting).

We furthermore state in the text that we see multiple local dips in Fig. 3f:

Conversely, the metasurface-stabilized cavity shows local dips for the longitudinal modes with the indices $q = 4$ and $q = 9, 10, 11$, see Figs. 3f, h. Our simulations reproduce these transmission dips for the longitudinal modes with the indices $q = 4$ and $q = 8, 9, 10$ (see Figs. 2a, c). We attribute the small offset of the longitudinal mode numbers to fabrication effects that cause a slightly decreased metasurface effective radius of curvature (see below).

and distinguish before discussing Fig. 4e, f, g, h:

For Hermite-Gaussian modes, a single metasurface design with a fixed radius of curvature can stabilize many longitudinal modes (with the cavity length limited by the stability criterion). This works because a mode's propagation length-dependent radius of curvature can be counteracted by a change in its minimum beam waist, such that its wavefront fits the metasurface's effective radius of curvature.

(3) In Fig. 4e, the first five modes cannot exhibit symmetric Lorentzian line shapes, which is similar with that without metasurface (Fig. 3c). It seems that the introduced metasurface does not work. Can the author explain it in detail and give the cavity length-dependent transmission of a microcavity without metasurface?

The reviewer is right that the metasurface does not stabilize modes that are not close to the design cavity length. This is expected behavior, as the H-shaped (and most other complicated) mode is not able to adapt to a changing cavity length by changing its waist size, as a Hermite-Gaussian mode would.

We have added a new inset to Fig. 4e to highlight the changing mode profile with cavity length: the new inset shows the transverse mode profile of the locally enhanced longitudinal mode close to 4.5 μm cavity length (see black dashed arrow) in a cavity stabilized by the metasurface enhancing the H-shaped mode, which is excited with a light beam with Gaussian profile. Comparison with the longitudinal mode close to 10 μm cavity length, for which the metasurface is designed, shows that the H-shape is only enhanced close to the design cavity length.

We have furthermore added new simulations and data plots to highlight the importance of the metasurface stabilization and the difference between a cavity with and without a metasurface (i.e., a Fabry-Perot cavity):

The new Fig. 4f shows the transmission of the metasurface-stabilized cavity, excited with an H-shaped beam matched to the designed transverse cavity mode profile. We also see the local enhancement at the design cavity length. The inset shows the transverse mode profile at this cavity length, which closely resembles the desired H-shape.

As demanded, the new Fig. 4g shows the transmission of an empty (i.e., Fabry-Perot) cavity, excited with the same H-shaped beam as used in Fig. 4f. The inset shows the transverse mode profile at the design cavity length. Even though the exciting beam is H-shaped, the transverse mode profile does not resemble the desired H-shape, reiterating the necessity of the metasurface.

Moreover, the resonance transmission peak shapes close to the design cavity length are shown in the new Fig. 4h. Whereas the metasurface-stabilized cavity shows a Lorentzian transmission peak shape, the asymmetric transmission peak shape of the Fabry-Perot cavity indicates it is unstable.

We have also extended our discussion of the cavity length-dependent peak transmission to clarify this behavior:

For Hermite-Gaussian modes, a single metasurface design with a fixed radius of curvature can stabilize many longitudinal modes (with the cavity length limited by the stability criterion). This works because a mode's propagation length-dependent radius of curvature can be counteracted by a change in its minimum beam waist, such that its wavefront fits the metasurface's effective radius of curvature.

This does not apply when the desired mode is a hologram. In this situation, transmission peaks for cavity lengths that are much shorter or much longer than the design value show asymmetric mode profiles, as is characteristic of unstable modes. The transmission peak close to the design cavity length shows a Lorentzian profile (see Fig. 4h). This is further illustrated by the transverse mode profiles at resonance: close to the design cavity length (see Fig. 4d), the desired H-shape is produced. Off of the design cavity length (see inset in Fig. 4e), the mode profile does not resemble the desired H-shape. Another indication of the length-dependent stability is that the cavity length-dependent resonance transmission (see Fig. 4e, f) shows a local enhancement close to the design cavity length. An empty Fabry-Perot cavity, even if excited with an H-shaped mode, does not show a local enhancement at the design cavity length (see Fig. 4g), nor does the corresponding enhanced field have an H-shaped profile (see inset in Fig. 4g). This reiterates the necessity of metasurfaces to enhance modes with complicated transverse profiles. Whereas the metasurface guarantees enhancement of the longitudinal mode at the design cavity length, it is not designed to suppress other longitudinal modes. Therefore, other local maxima, especially for short cavity lengths that

limit the transverse mode spreading per cavity round trip, can occur (see, e.g., Fig. 4e black dashed arrow), albeit with uncontrolled transverse mode profiles.

(4) It would be great if the metasurface design of the asymmetric dot-pattern is given.

We added the metasurface design of the asymmetric dot pattern in a new panel Fig. 4j.

Reviewer #4 (Remarks to the Author):

The manuscript is dealing with the experimental results of the stabilized optical microcavities whose one commercially available DBR mirror's surface is modified with metasurface of nano pillars. The idea and results are interesting and have significant potentials for real usage. [... see (1), (2) below ...] The content is worth publishing in Nature Communications.

We thank the reviewer for the very positive review and the suggestion for publication in Nature Communications.

(1) The durability for real usage is also interesting. If authors have any knowledge for this question, please give us authors' comments.

We agree and have expanded on the durability in the manuscript:

Both, chemical and mechanical durability are experimentally verified, as in the final fabrication step (see methods), the device is immersed in bubbling Piranha solution. To increase the durability even more, i.e., for applications where the metasurface will likely touch hard objects, the Silicon pillars can be protected fully by incorporating them in a fused Silica matrix. We then placed the manufactured device opposite to a planar DBR and measured the wavelength and cavity length-dependent transmission of the resulting cavity for a focused incident light beam (numerical aperture $NA \approx 0.3$, wavelength 1520 – 1580 nm, see methods for details). For the measurements, we oscillated the cavity length more than 100.000 times using a piezoelectric stage. Fig. 3e shows a scanning electron microscopy picture of a final device after its measurement.

(2) The fabrication method for metasurface is presented in Methods part, but no illustration. Some illustration for fabrication of the metasurface on DBR mirror is very useful for readers to understand how we fabricate the metasurface on DBR mirrors.

We agree and have added a new Supplementary Figure 1 detailing the fabrication process. We also included a reference to the new figure in the main text:

Using top-down processing (see methods and Supplementary Fig. 1), we fabricated such metasurfaces on top of a commercially available Silica/Titania DBR terminated with a Silica layer (reflectivity >99.5%).

And to the fabrication section of the methods:

The fabrication process is summarized in Supplementary Fig. 1.

Reviewer #1 (Remarks to the Author):

The added supplementary Figure 4 and discussions provide some fundamental information to the engineering design. With this, I remove my concerns.

Reviewer #2 (Remarks to the Author):

The Authors have addressed all my comments and suggestions. The new additions improve these aspects of the manuscript and therefore I consider that the manuscript can be published as it is in Nature Communications.

Reviewer #3 (Remarks to the Author):

The comments of this manuscript have been carefully addressed. I think the manuscript should be considered for publication in Nature Communications.

Reviewer #4 (Remarks to the Author):

Final Report on Manuscript entitled "Metasurface-Stabilized Optical Microcavities" by Dr Ossiander and colleagues (Manuscript Number: NCOMMS-22-32850), to be published in Nature Communications

Authors sincerely and properly responded all comments by Reviewers. I recommend the manuscript can be accepted for publication.

Specific comments and answers to the reviewers

In the following, we give a one-to-one response (printed in blue) to the individual reviewer's comments which are printed verbatim and in their entirety in green.

Reviewer #1 (Remarks to the Author):

The added supplementary Figure 4 and discussions provide some fundamental information to the engineering design. With this, I remove my concerns.

We thank the reviewer for removing her/his concern.

Reviewer #2 (Remarks to the Author):

The Authors have addressed all my comments and suggestions. The new additions improve these aspects of the manuscript and therefore I consider that the manuscript can be published as it is in Nature Communications.

We thank the reviewer for the positive review and the suggestion for publication in Nature Communications.

Reviewer #3 (Remarks to the Author):

The comments of this manuscript have been carefully addressed. I think the manuscript should be considered for publication in Nature Communications.

We thank the reviewer for the positive review and the suggestion for publication in Nature Communications.

Reviewer #4 (Remarks to the Author):

Final Report on Manuscript entitled "Metasurface-Stabilized Optical Microcavities" by Dr Ossiander and colleagues (Manuscript Number: NCOMMS-22-32850), to be published in Nature Communications

Authors sincerely and properly responded all comments by Reviewers. I recommend the manuscript can be accepted for publication.

We thank the reviewer for the positive review and the suggestion for publication in Nature Communications.